# ResNet with one-neuron hidden layers is a Universal Approximator

**Hongzhou Lin**
MIT
Cambridge, MA 02139
hongzhou@mit.edu

**Stefanie Jegelka**
MIT
Cambridge, MA 02139
stefje@mit.edu

## Abstract

We demonstrate that a very deep ResNet with stacked modules that have one neuron per hidden layer and ReLU activation functions can uniformly approximate any Lebesgue integrable function in $d$ dimensions, i.e. $\ell_1(\mathbb{R}^d)$. Due to the identity mapping inherent to ResNets, our network has alternating layers of dimension one and $d$. This stands in sharp contrast to fully connected networks, which are not universal approximators if their width is the input dimension $d$ [21, 11]. Hence, our result implies an increase in representational power for narrow deep networks by the ResNet architecture.

## 1 Introduction

Deep neural networks are central to many recent successes of machine learning, including applications such as computer vision, natural language processing, or reinforcement learning. A common trend in deep learning has been to construct larger and deeper networks, starting from the pioneer convolutional network LeNet [19], to networks with tens of layers such as AlexNet [17] or VGG-Net [28], or recent architectures like GoogLeNet/Inception [30] or ResNet [13, 14], which may contain hundreds or thousands of layers. A typical observation is that deeper networks offer better performance. This phenomenon, at least on the training set, supports the intuition that a deeper network should have more capacity to approximate the target function, and leads to a question that has received increasing interest in the theory of deep learning: can all functions that we may care about be approximated well by a sufficiently large and deep network? In this work, we address this important question for the popular ResNet architecture.

The question of representational power of neural networks has been answered in different forms. Results in the late eighties showed that a network with a single hidden layer can approximate any continuous function with compact support to arbitrary accuracy, when the width goes to infinity [7, 15, 10, 18]. This result is referred to as the *universal approximation theorem*. Analogous to the classical Stone-Weierstrass theorem on polynomials or the convergence theorem on Fourier series, this theorem implies that the family of neural networks are universal approximators: we can apply neural networks to approximate any continuous function and the accuracy improves as we add more neurons in the width. More importantly, the coefficients in the network can be efficiently learned via back-propagation, providing an explicit representation of the approximation.

This classical universal approximation theorem completely relies on the power of the width increasing to infinity, i.e., "fat" networks. Current "tall" deep learning models, however, are not captured by this setting. Consequently, theoretically analyzing the benefit of depth has gained much attention in the recent literature [31, 6, 9, 32, 23, 20, 25]. The main focus of these papers is to provide examples of functions that can be efficiently represented by a deep network but are hard to represent by shallow networks. These examples require exponentially many neurons in a shallow network to achieve the same approximation accuracy as a deep network with only a polynomial or linear number of

neurons. Yet, these specific examples do not imply that *all* shallow networks can be represented by deep networks, leading to an important question:

*If the number of neurons in each layer is bounded, does universal approximation hold when the depth goes to infinity?*

This question has recently been studied by [21, 11] for fully connected networks with ReLU activation functions: if each hidden layer has at least $d + 1$ neurons, where $d$ is the dimension of the input space, the universal approximation theorem holds as the depth goes to infinity. If, however, at most $d$ neurons can be used in each hidden layer, then universal approximation is impossible even with infinite depth.

In practice, other architectures have been developed to improve empirical results. A popular example is ResNet [13, 14], which includes an identity mapping in addition to each layer. A first step towards a better theoretical understanding of those empirically successful models is to ask how the above question extends to them. Do the architecture variations make a difference theoretically? Due to the identity mapping, for ResNet, the width of the network remains the same as the input dimension. For a formal analysis, we stack modules of the form shown in Figure 1, and analyze how small the hidden green layers can be. The resulting width of $d$ (blue) or even less (green) stands in sharp contrast with the negative result for width $d$ for fully connected networks in [21, 11]. Indeed, our empirical illustrations in Section 2 demonstrate that, empirically, significant differences in the representational power of narrow ResNets versus narrow fully connected networks can be observed. Our theoretical results confirm those observations.

Hardt and Ma [12] show that ResNet enjoys universal finite-sample expressive power, i.e., ResNet can represent any classifier on any finite sample perfectly. This positive result in the discrete setting motivates our work. Their proof, however, relies on the fact that samples are "far" from each other and hence cannot be used in the setting of full functions in continuous space.

**Contributions.** The main contribution of this paper is to show that ResNet with one single neuron per hidden layer is enough to provide universal approximation as the depth goes to infinity. More precisely, we show that for any Lebesgue-integrable[1] function $f : \mathbb{R}^d \to \mathbb{R}$, for any $\epsilon > 0$, there exists a ResNet $R$ with ReLU activation and one neuron per hidden layer such that

$$\int_{\mathbb{R}^d} |f(x) - R(x)| dx \leq \epsilon.$$

This result implies that, compared to fully connected networks, the identity mapping of ResNet indeed adds representational power for tall networks.

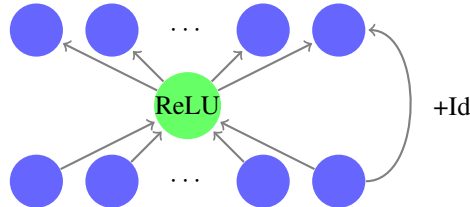

Figure 1: The basic residual block with one neuron per hidden layer.

The ResNet in our construction is built by stacking residual blocks of the form illustrated in Figure 1, with one neuron in the hidden layer. A basic residual block consists of two linear mappings and a single ReLU activation [12, 13]. More formally, it is a function $\mathcal{T}_{U,V,u}$ from $\mathbb{R}^d$ to $\mathbb{R}^d$ defined by

$$\mathcal{T}_{U,V,u}(x) = V\text{ReLU}(Ux + u),$$

where $U \in \mathbb{R}^{1 \times d}$, $V \in \mathbb{R}^{d \times 1}$, $u \in \mathbb{R}$ and the ReLU activation function is defined by

$$\text{ReLU}(x) = \max(x, 0) = [x]_+. \tag{1}$$

After performing the nonlinear transformation, we add the identity to form the input of the next layer. The resulting ResNet is a combination of several basic residual blocks and a final linear output layer:

$$R(x) = \mathcal{L} \circ (Id + \mathcal{T}_N) \circ (Id + \mathcal{T}_{N-1}) \circ \cdots \circ (Id + \mathcal{T}_0)(x),$$

where $\mathcal{L} : \mathbb{R}^d \to \mathbb{R}$ is a linear operator and $\mathcal{T}_i$ are basic one-neuron residual blocks.

Unlike the original architecture [13], we do not include any convolutional layers, max pooling or batch normalization; the above simplified architecture turns out to be sufficient for universal approximation.

## 2  A motivating example

We begin by empirically exploring the difference between narrow fully connected networks, with $d$ neurons per hidden layer, and ResNet via a simple example: classifying the unit ball in the plane.

The training set consists of randomly generated samples $(z_i, y_i)_{i=1\cdots n} \in \mathbb{R}^2 \times \{-1, 1\}$ with

$$y_i = \begin{cases} 1 & \text{if } \|z_i\|_2 \leq 1; \\ -1 & \text{if } 2 \leq \|z_i\|_2 \leq 3. \end{cases}$$

We artificially create a margin between positive and negative samples to make the classification task easier. As loss, we use the logistic loss $\frac{1}{n} \sum \log(1 + e^{-y_i \hat{y}_i})$, where $\hat{y}_i = f_{\mathcal{N}}(z_i)$ is the output of the network on the $i$-th sample. After training, we illustrate the learned decision boundaries of the networks for various depths. Ideally, we would expect the decision boundaries of our models to be close to the true distribution, i.e., the unit ball.

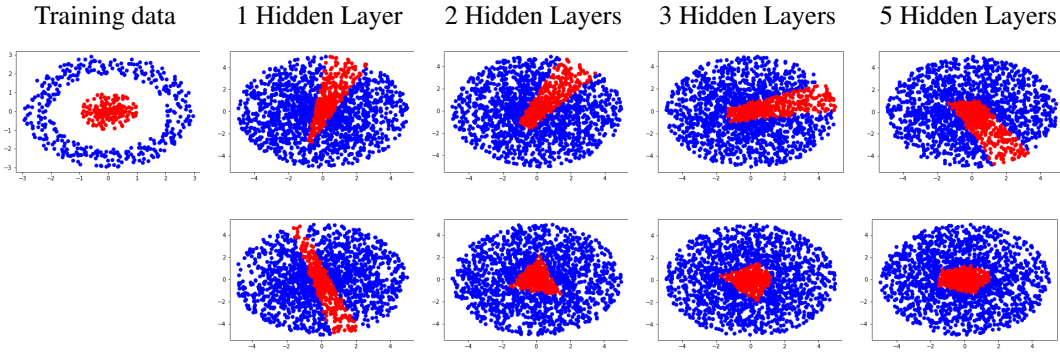

Figure 2: Decision boundaries obtained by training fully connected networks with width $d = 2$ per hidden layer (top row) and ResNet (bottom row) with one neuron in the hidden layers on the unit ball classification problem. The fully connected networks fail to capture the true function, in line with the theory stating that width $d$ is too narrow for universal approximation. ResNet in contrast approximates the function well, empirically supporting our theoretical results.

Figure 2 shows the results. For the fully connected networks (top row), the learned decision boundaries have roughly the same shape for different depths: the approximation quality seems to not improve with increasing depth. While one may be inclined to argue that this is due to local optimality, our observation agrees with the results in [21]:

**Proposition 2.1.** *Let $f_{\mathcal{N}} : \mathbb{R}^d \to \mathbb{R}$ be the function defined by a fully connected network $\mathcal{N}$ with ReLU activation. Denote by $P = \{x \in \mathbb{R}^d \,|\, f_{\mathcal{N}}(x) > 0\}$ the positive level set of $f_{\mathcal{N}}$. If each hidden layer of $\mathcal{N}$ has at most $d$ neurons, then*

$$\lambda(P) = 0 \quad \text{or} \quad \lambda(P) = +\infty, \quad \text{where } \lambda \text{ denotes the Lebesgue measure.}$$

*In other words, the non-trivial level set of a narrow fully connected network is always unbounded.*

The proof is a direct application of Theorem 2 of [21], see Appendix E. Thus, even when the depth goes to infinity, a narrow fully connected network can never approximate a bounded region. Here we only show the case $d = 2$ because we can easily visualize the data; the same observation will still hold in higher dimensions. An even stronger, recent result states that any connected component of the decision boundaries obtained by a narrow fully connected network is unbounded [3].

The decision boundaries for ResNet appear strikingly different: despite the even narrower width of one, from 2 hidden layers onwards, the ResNet represents the indicator of a bounded region.

With increasing depth, the decision boundary seems to converge to the unit ball, implying that Proposition 2.1 cannot hold for ResNet. These observations motivate the universal approximation theorem that we will show in the next section.

# 3 Universal approximation theorem

In this section, we present the universal approximation theorem for ResNet with one-neuron hidden layers. We sketch the proof in the one-dimensional case; the induction for higher dimensions relies on similar ideas and may be found in the appendix.

**Theorem 3.1** (**Universal Approximation of ResNet**). *For any $d \in \mathbb{N}$, the family of ResNet with one-neuron hidden layers and ReLU activation function can universally approximate any $f \in \ell_1(\mathbb{R}^d)$. In other words, for any $\epsilon > 0$, there is a ResNet $R$ with finitely many layers such that*

$$\int_{\mathbb{R}^d} |f(x) - R(x)| dx \leq \epsilon.$$

**Outline of the proof.** The proof starts with a well-known fact: the class of piecewise constant functions with compact support and finitely many discontinuities is dense in $\ell_1(\mathbb{R}^d)$. Thus it suffices to approximate any piecewise constant function. Given a piecewise constant function, we first construct a grid "indicator" function on its support, as shown in Figure 4. This function is similar to an indicator function in the sense that it vanishes outside the support, but, instead of being constantly equal to one, a grid indicator function takes different constant values on different grid cells, see Definition B.3 for a formal definition. The property of having different function values creates a "fingerprint" on each grid cell, which will help to distinguish them. Then, we divide the space into different level sets, such that one level set contains exactly one grid cell. Finally, we fit the function value on each grid cell, cell by cell.

**Sketch of the proof when $d = 1$.** We start with the one-dimensional case, which is central to our construction. As mentioned above, it is sufficient to approximate piecewise constant functions. Given a piecewise constant function $h$, there is a subdivision $-\infty < a_0 < a_1 < \cdots < a_M < +\infty$ such that

$$h(x) = \sum_{k=1}^{M} h_k \mathbb{1}_{x \in [a_{k-1}, a_k)},$$

where $h_k$ is the constant value on the $k$-th subdivision $I_k = [a_{k-1}, a_k)$. We will approximate $h$ via trapezoid functions of the following form, shown in Figure 3.

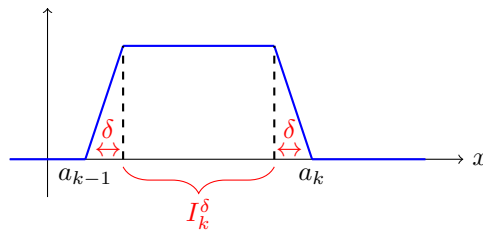

Figure 3: A trapezoid function, which is a continuous approximation of the indicator function. The parameter $\delta$ measures the quality of the approximation.

A trapezoid function is a simple continuous approximation of the indicator function. It is constant on the segment $I_k^\delta = [a_{k-1} + \delta, a_k - \delta]$ and linear in the $\delta$-tolerant region $I_k \setminus I_k^\delta$. As $\delta$ goes to zero, the trapezoid function tends point-wise to the indicator function.

A natural idea to approximate $h$ is to construct a trapezoid function on each subdivision $I_k$ and to then sum them up. This is the main strategy used in [21, 11] to show a universal approximation theorem for fully connected networks with width at least $d + 1$. However, this strategy is not applicable for the ResNet structure because the summation requires memory of past components, and hence requires additional units in every layer. The width constraint of ResNet makes the difference here.

In contrast, we construct our approximation in a sequential way: we build the components of the trapezoid function one after another. With this sequential construction, we can only build increasing

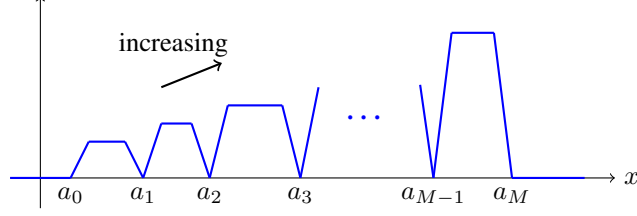

Figure 4: An increasing trapezoid function, which is a special case of grid indicator function when $d = 1$, is trapezoidal on each subdivision with increasing constant value from left to right.

trapezoid functions as shown in Figure 4. Such functions are trapezoidal on each subdivision $I_k$ and the constant value on $I_k^\delta$ increases when $k$ grows. The construction relies on the following basic operations:

**Proposition 3.2** (**Basic operations**). *The following operations are realizable by a single basic residual block of ResNet with one neuron:*

*(a)* ***Shifting by a constant:*** $R^+ = R + c$ *for any* $c \in \mathbb{R}$;

*(b)* ***Min or Max with a constant:*** $R^+ = \min\{R, c\}$ *or* $R^+ = \max\{R, c\}$ *for any* $c \in \mathbb{R}$;

*(c)* ***Min or Max with a linear transformation:*** $R^+ = \min\{R, \alpha R + \beta\}$ *(or* $\max$*) for any* $\alpha, \beta \in \mathbb{R}$;

*where $R$ represents the input layer in the basic residual block and $R^+$ the output layer.*

Geometrically, operation (a) allows us to shift the function by a constant; operation (b) allows us to remove the level set $\{R \geq c\}$ or $\{R \leq c\}$ and operation (c) can be used to adjust the slope. With these basic operations at hand, we construct the increasing trapezoid function by induction on the subdivisions. For any $m \in [0, M]$, we construct a function $R_m$ satisfying

C1. $R_m = 0$ on $(-\infty, a_0]$;

C2. $R_m$ is a trapezoid function on each $I_k$, for any $k = 1, \cdots, m$;

C3. $R_m = (k+1)\|h\|_\infty$ on $I_k^\delta = [a_{k-1} + \delta, a_k - \delta]$ for any $k = 1, \cdots, m$;

C4. $R_m$ is bounded on $(-\infty, a_m]$ by $0 \leq R_m \leq (m+1)\|h\|_\infty$;

C5. $R_m(x) = -\frac{(m+1)\|h\|_\infty}{\delta}(x - a_m)$ if $x \in [a_m, +\infty)$;

where $\|h\|_\infty = \max\limits_{k=1\cdots M} |h_k|$ is the infinity norm and $\delta > 0$ measures the quality of the approximation. An illustration of $R_m$ is shown in Figure 5. On the first $m$ subdivisions, $R_m$ is the restriction of the desired increasing trapezoid function. On $[a_m, +\infty)$, the function $R_m$ is a very steep linear function with negative slope that enables the construction of next subdivision.

Given $R_m$, we sequentially stack three residual blocks to build $R_{m+1}$:

- $R_m^+ = \max\left\{R_m, -\left(1 + \frac{1}{m+1}\right)R_m\right\}$;

- $R_m^{++} = \min\left\{R_m^+, -R_m^+ + \frac{(m+2)\|h\|_\infty}{\delta}(a_{m+1} - a_m)\right\}$;

- $R_{m+1} = \min\{R_m^{++}, (m+2)\|h\|_\infty\}$.

Figure 5 illustrates the effect of these blocks: the first operation flips the linear part on $[a_m, +\infty)$ by adjusting the slope, the second operation folds the linear function in the middle of $[a_m, a_{m+1}]$, and finally we cut off the peak at the appropriate level $(m+2)\|h\|_\infty$.

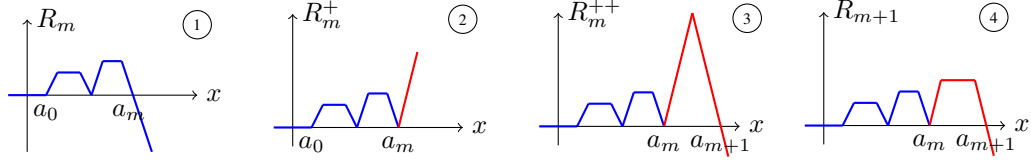

Figure 5: The construction of $R_{m+1}$ based on $R_m$. We build the next trapezoid function (red) and keep the previous ones (blue) unchanged.

An important consideration is that we need to keep the function on previous subdivisions unchanged while building the next trapezoid function. We achieve this by increasing the function values. The different values will be the basis for adjusting the function value in each subdivision to the final value of the target function we want to approximate. Before proceeding with the adjustment, we remark that $R_M$ goes to $-\infty$ as $x \to \infty$. This negative "tail" is easily removed by performing a cut-off operation via the max operator. This gives us the desired increasing trapezoid function $R_M^*$.

To adjust the function values on the intervals $I_k^\delta$, we identify the $I_k^\delta$ via the level sets of $R_M^*$. This works because, by construction, $R_M^* = (k+1)\|h\|_\infty$ on $I_k^\delta$. More precisely, we define the level sets $L_k = \{k\|h\|_\infty < R_M^* \le (k+1)\|h\|_\infty\}$ (for $k = 0, \cdots, M$) and adjust them one by one from highest to lowest value: for any $k = M, \cdots, 1$, we sequentially build

$$R_{k-1}^* = R_k^* + \frac{h_k - (k+1)\|h\|_\infty}{\|h\|_\infty}[R_k^* - k\|h\|_\infty]_+. \tag{2}$$

An illustration of the $R_k^*$ is shown in Figure 6.

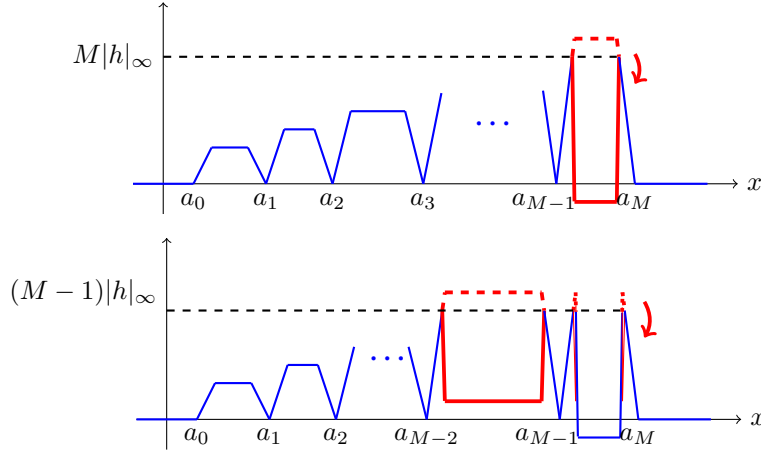

Figure 6: An illustration of the function adjustment procedure applied to the top level sets. At each step, we adjust one $I_k^\delta$ to the desired function value $h_k$.

In particular, the first step from $R_M^*$ to $R_{M-1}^*$ only scales the top level set because the ReLU activation $[R_M - M\|h\|_\infty]_+$ is active if and only if $x \in L_M$. The coefficients are appropriately selected such that after the scaling, the constant in $I_M^\delta$ matches $h_M$. Hence, we have

$$R_{M-1}^* = \begin{cases} h_M & \text{if } x \in I_M^\delta \subset L_M; \\ R_M^* & \text{if } x \notin L_M. \end{cases}$$

Next, we set the second largest level set to $h_{M-1}$, and so on. As a result, the function $R_0^*$, obtained after rescaling all the level sets is the desired approximation of the piecewise constant function $h$. Concretely, we show that $R_0^*$ satisfies

- $R_0^* = 0$ on $(-\infty, a_0]$ and $[a_M, +\infty)$;
- $R_0^* = h_k$ on $I_k^\delta = [a_{k-1} + \delta, a_k - \delta]$ for any $k = 1, \cdots, M$;

- $R_0^*$ is bounded with $-\|h\|_\infty \le R_0^* \le \|h\|_\infty$.

The detailed proof is deferred to Appendix B. Importantly, our construction is valid for any small enough $\delta$ satisfying $0 < 2\delta < \min_{k=1,\cdots,M}\{a_k - a_{k-1}\}$. Hence, the approximation error, which is bounded by

$$\int_{\mathbb{R}} |R_0^*(x) - h(x)|dx \le 4M\delta\|h\|_\infty,$$

can be made arbitrarily small by taking $\delta$ to 0. This completes the proof.

**Extension to higher dimensions.** The last step of the one-dimensional construction is performed by sliding through all the grid cells and adjusting the function value sequentially. This procedure can be done regardless of the dimension. Therefore, it suffices to build a $d$-dimensional grid indicator function, which generalizes the notion of increasing trapezoid function into high dimensional space (Definition B.3 in the appendix).

We perform an induction over dimensions. The main idea is to sum up an appropriate one-dimensional grid indicator function and an appropriate $d-1$ dimensional grid indicator function, as illustrated in Figure 7.

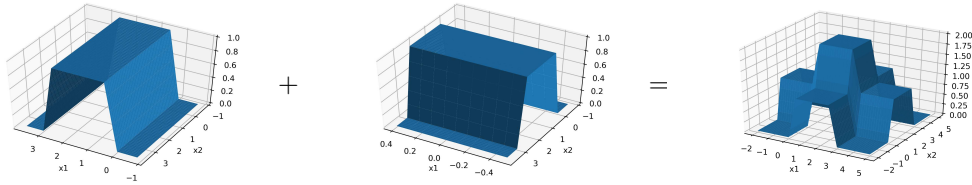

Figure 7: One dimensional grid indicator functions on the first (left) and second (middle) coordinate. Both functions can be constructed independently by our one hidden unit ResNet.

The summation gives the desired shape inside each grid cell. However, some regions outside the grid cells are also raised, but were supposed to be zero. We address this issue via another, separate level set property: there is a threshold $T$ such that a) the function value inside each $I_k^\delta$ is larger than $T$; b) the function values outside the grid cells are smaller than $T$. Therefore, the desired grid indicator function can be obtained by performing a max operator with the threshold $T$, i.e., cutting off the smaller values and setting them to zero (see Appendix C).

**Number of neurons/layers.** A straightforward consequence of our construction is that we can approximate any piecewise constant function to arbitrary accuracy with a ResNet of $O(\text{number of grid cells})$ hidden units/layers. The most space-consuming procedure is the function adjustment that requires going through each of the grid cells one by one. This procedure however can be parallelized if we allow more hidden units per layer.

Deriving an exact relationship between the original target function $f$ and the required number of grid cells is nontrivial and highly dependent on characteristics of $f$. In particular, when the function $f$ is continuous, this number is related to the modulus of continuity of $f$ defined by

$$\omega_K(r) = \max_{x,y \in K, \|x-y\| \le r} |f(x) - f(y)|,$$

where $K$ is any compact set and $r$ represents the radius of the discretization. Given a desired approximation accuracy $\epsilon$, we need to

- first, determine a compact set $K$ such that $\int_{\mathbb{R}^d \setminus K} |f| \le \epsilon$ and restrict $f$ to $K$;
- second, determine $r$ such that $\omega_K(r) \le \epsilon/\mathrm{Vol}(K)$.

Then, the number of grid cells is $O(1/r^d)$. This dependence is suboptimal in the exponent, and it may be possible to improve it using a similar strategy as [34]. Also, by imposing stronger smoothness assumptions, this number may be reducible dramatically [2, 22, 33]. These improvements are not the main focus of this paper, and we leave them for future work.

# 4   Discussion and concluding remarks

In this paper, we have shown a universal approximation theorem for the ResNet structure with one unit per hidden layer. This result stands in contrast to recent results on fully connected networks, for which universal approximation fails with width $d$ or less. To conclude, we add some final remarks and implications.

**ResNet vs Fully connected networks.**   While we achieve universal approximation with only one hidden neuron in each basic residual block, one may argue that the structure of ResNet still passes the identity to the next layer. This identity map could be counted as $d$ hidden units, resulting in a total of $d + 1$ hidden unites per residual block, and could be viewed as making the network a width $(d + 1)$ fully connected network. But, even from this angle, ResNet corresponds to a compressed or sparse version of a fully connected network. In particular, a width $(d + 1)$ fully connected network has $O(d^2)$ connections per layer, whereas only $O(d)$ connections are present in ResNet thanks to the identity map. This "overparametrization" of fully connected networks may be a partial explanation why dropout [29] has been observed to be beneficial for such networks. By the same argument, our result implies that width $(d + 1)$ fully connected networks are universal approximators, which is the minimum width needed [11]. A detailed construction may be found in Appendix F.

**Why does universal approximation matter?**   As shown in Section 2, a width $d$ fully connected network can never approximate a compact decision boundary even if we allow infinite depth. However, in high dimensional space, it is very hard to visualize and check the obtained decision boundary. The universal approximation theorem then provides a sanity check, and ensures that, in principle, we are able to capture any desired decision boundary.

**Training efficiency.**   The universal approximation theorem only guarantees the possibility of approximating any desired function, but it does not guarantee that we will actually find it in practice by running SGD or any other optimization algorithm. Understanding the efficiency of training may require a better understanding of the optimization landscape, a topic of recent attention [5, 16, 24, 26, 8, 35, 27].

Here, we try to provide a slightly different angle. By our theory, ResNet with one-neuron hidden layers is already a universal approximator. In other words, a ResNet with multiple units per layer is in some sense an over-parametrization of the model, and over-parametrization has been observed to benefit optimization [36, 4, 1]. This might be one reason why training a very deep ResNet is "easier" than training a fully connected network. A more rigorous analysis is an interesting direction for future work.

**Generalization.**   Since a universal approximator is able to fit any function, one might expect it to overfit very easily. Yet, it is commonly observed that deep networks generalize surprisingly well on the test set. The explanation of this phenomenon is orthogonal to our paper, however, knowing the universal approximation capability is an important building block of such a theory. Moreover, the above-mentioned "over-parametrization" implied by our results may play a role too.

To conclude, we have shown a universal approximation theorem for ResNet with one-neuron hidden layers. This theoretically distinguishes them from fully connected networks. To some extent, our construction also theoretically motivates the current practice of going deeper and deeper in the ResNet architecture.

### Acknowledgements

We would like to thank Jeffery Z. HaoChen for useful feedback and suggestions for this paper. This research was supported by The Defense Advanced Research Projects Agency (grant number YFA17 N66001-17-1-4039). The views, opinions, and/or findings contained in this article are those of the author and should not be interpreted as representing the official views or policies, either expressed or implied, of the Defense Advanced Research Projects Agency or the Department of Defense.

## Footnotes

[1] A function $f$ is Lebesgue-integrable if $\int_{\mathbb{R}^d} |f(x)| dx < \infty$.

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
