[Supplementary Material · resnet_final_supplementary.pdf]

This supplementary material is devoted to the theoretical proof of the universal approximation theorem of ResNet. We start with the one dimensional case and some basic operations, then we extend the result to high dimension by induction.

## A    Notation and preliminaries

In this section, we set up the notation and prepare some tools towards the universal approximation theorem. We first define the class of piecewise constant functions with compact support and finitely many discontinuities.

**Definition A.1.** *A function $h : \mathbb{R}^d \to \mathbb{R}$ is piecewise constant with compact support and finitely many discontinuities if we can partition the space into finitely many grid cells such that $h$ vanishes outside the grid and is constant inside each grid cell. More precisely, for any coordinate $i \in [1, d]$, there is a subdivision and $a_0^i < \cdots < a_{M_i}^i$ such that*

1. *$h = 0$ outside $I = [a_0^1, a_{M_1}^1] \times [a_0^2, a_{M_2}^2] \times \cdots \times [a_0^d, a_{M_d}^d]$;*

2. *$h$ is constant on each small cube $[a_{i_1}^1, a_{i_1+1}^1] \times [a_{i_2}^2, a_{i_2+1}^2] \times \cdots \times [a_{i_d}^d, a_{i_d+1}^d]$.*

*We denote the family of piecewise constant function with compact support and finitely many discontinuities by $PC(\mathbb{R}^d)$. For simplicity, in the following, we refer to them as piecewise constant functions.*

**Theorem A.2.** *The class of piecewise constant functions is dense in $\ell_1(\mathbb{R}^d)$.*

Theorem A.2 is a well known result, which can be directly derived from the definition of Lebesgue measure. As a consequence, it is sufficient to prove that any piecewise constant function can be approximated by a ResNet up to arbitrary accuracy. We start by providing some basic operations of one-neuron residual blocks.

**Proposition A.3** (**Basic operations**)**.** *The following operations are realizable by a single basic residual block of ResNet with one hidden neuron:*

*(a)* **Shifting by a constant:** *$R^+ = R + c$ for any $c \in \mathbb{R}$;*

*(b)* **Min or Max with a constant:** *$R^+ = \min\{R, c\}$ or $R^+ = \max\{R, c\}$ for any $c \in \mathbb{R}$;*

*(c)* **Min or Max with a linear transformation:** *$R^+ = \min\{R, \alpha R + \beta\}$ (or max) for any $\alpha, \beta \in \mathbb{R}$;*

*where $R$ represents the input layer in the basic residual block and $R^+$ the output layer.*

Figure 8: Basic residual block with one neuron per hidden layer in dimension one.

*Proof.* It is sufficient to prove (c) since it implies (a) and (b). Indeed, given $\alpha, \beta \in \mathbb{R}$, for any input $R$,

$$\max\{R, \alpha R + \beta\} = R + [(\alpha - 1)R + \beta]_+$$
$$\text{and} \quad \min\{R, \alpha R + \beta\} = R - [(1 - \alpha)R - \beta]_+,$$

which are both realizable by basic residual blocks.    $\square$

These basic operations will be extensively used in the upcoming construction. Intuitively, operation (a) allows us to shift the function by a constant value; operation (b) allows us to cut off the level set $\{R \geq c\}$ or $\{R \leq c\}$ and operation (c), which is more complex, can be used to adjust the slope of the function.

# B    Warm Up: One-Dimensional case

We start with the one-dimensional case, which is central to our construction. As mentioned above, it is sufficient to approximate piecewise constant functions. Given a piecewise constant function $h$, there is a subdivision $-\infty < a_0 < a_1 < \cdots < a_M < +\infty$ such that

$$h(x) = \sum_{k=1}^{M} h_k \mathbb{1}_{x \in [a_{k-1}, a_k)},$$

where $h_k$ is the constant value on the $k$-th subdivision $I_k = [a_{k-1}, a_k)$. We will approximate $h$ via trapezoid functions, which are continuous approximations of indicator functions, as shown in Figure 9.

Figure 9: A trapezoid function, which is a continuous approximation of the indicator function. The parameter $\delta$ measures the quality of the approximation.

A trapezoid function is constant on the segment $I_k^\delta = [a_{k-1} + \delta, a_k - \delta]$ and linear in the $\delta$-tolerant region $I_k \backslash I_k^\delta$. As $\delta$ goes to zero, the trapezoid function tends point-wise to the indicator function.

A natural idea to approximate a piecewise constant function is to construct a trapezoid function on each subdivision $I_k$ and to then sum them up. This is the main strategy used in [21, 11] to show a universal approximation theorem for fully connected networks with width at least $d + 1$. However, this strategy is not applicable for the ResNet structure because the summation requires memory of past components, and hence requires additional units in every layer. The width constraint of ResNet makes the difference here.

**Proposition B.1.** *Given a piecewise constant function $h$, for any $\delta > 0$ satisfying $2\delta < \min_{k=1,\cdots,M}\{a_k - a_{k-1}\}$, there exists a ResNet $R$ such that*

- $R(x) = 0$ *for $x \in (-\infty, a_0)$ and $x \in [a_M, +\infty)$;*

- $R(x) = h_k$ *for $x \in I_k^\delta = [a_{k-1} + \delta, a_k - \delta]$, for any $k = 1, \cdots, M$;*

- $R$ *is bounded with $-\|h\|_\infty \leq R \leq \|h\|_\infty$,*

*where $\|\cdot\|_\infty$ denotes the infinity norm.*

*Proof.* We construct our approximation in a sequential way, building the components of the trapezoid function one after another. With this sequential construction, we can only build increasing trapezoid functions as shown in Figure 10.

Figure 10: An increasing trapezoid function.

The construction is performed by induction on the subdivisions: for any $m \in [0, M]$, we build a ResNet $R_m$ satisfying

C1. $R_m = 0$ on $(-\infty, a_0]$;

C2. $R_m$ is a trapezoid function on each $I_k$, for any $k = 1, \cdots, m$;

C3. $R_m = (k+1)\|h\|_\infty$ on $I_k^\delta = [a_{k-1} + \delta, a_k - \delta]$ for any $k = 1, \cdots, m$;

C4. $R_m$ is bounded on $(-\infty, a_m]$ by $0 \le R_m \le (m+1)\|h\|_\infty$;

C5. $R_m(x) = -\frac{(m+1)\|h\|_\infty}{\delta}(x - a_m)$ if $x \in [a_m, +\infty)$.

**Initialization of the induction:** for $m = 0$, we start with the identity function and sequentially build

- $R^+ = \max\{x, a_0\} = x + [a_0 - x]_+$ (Cutting off $x \le a_0$);

- $R^{++} = R^+ - a_0$ (Shifting);

- $R_0 = R^{++} - \frac{(\|h\|_\infty + \delta)}{\delta}[R^{++}]_+$.

An illustration of the construction is shown in Figure 11 and the desired properties C1-C5 follow immediately from the construction.

Figure 11: An illustration of constructing the initial function $R_0$.

**Induction from $R_m$ to $R_{m+1}$:** given $R_m$, we stack three modules of one-neuron residual blocks on top of it to build $R_{m+1}$. More precisely, we use $R_m$ as input and sequentially perform

(a) $R_m^+ = R_m + \left(2 + \frac{1}{m+1}\right)[-R_m]_+$.

(b) $R_m^{++} = R_m^+ - 2\left[R_m^+ - (m+2)\|h\|_\infty \frac{a_{m+1} - a_m}{2\delta}\right]_+$.

(c) $R_{m+1} = \min\{R_m^{++}, (m+2)\|h\|_\infty\}$.

An illustration of the construction is shown in Figure 12.

Figure 12: The construction of $R_{m+1}$ based on $R_m$. We build the next trapezoid function (red) and keep the previous ones (blue) unchanged.

The first operation (a) adjusts the slope of the linear function on $[a_m, +\infty)$. This is possible since $R_m$ is positive on $(-\infty, a_m]$ and is negative on $[a_m, +\infty)$. The activation function $[-R_m]_+$ in

operation (a) is active if and only if $R_m \leq 0$, yielding

$$R_m^+ = \begin{cases} R_m & \text{if } x < a_m, \\ \frac{(m+2)\|h\|_\infty}{\delta}(x - a_m) & \text{if } x \in [a_m, +\infty). \end{cases}$$

The second operation (b) folds the linear function in the middle of $[a_m, a_{m+1}]$. This is possible since the slope on $[a_m, +\infty)$ is made steep enough such that the linear tail overpasses the largest value of previous subdivisions after an increment of size $\delta$. More precisely, we show that the ReLU activation function in (b) is active if and only if $x \geq \frac{a_m + a_{m+1}}{2}$.

• When $x < a_m$, $R_m^+ = R_m$, then by C4, $R_m$ is upper bounded by $(m+1)\|h\|_\infty$, yielding

$$R_m^+ = R_m \leq (m+1)\|h\|_\infty < (m+2)\|h\|_\infty \leq (m+2)\|h\|_\infty \frac{a_{m+1} - a_m}{2\delta},$$

where the last inequality holds since $2\delta < a_{m+1} - a_m$. Thus the ReLU activation function in operation (b) is not active when $x < a_m$, meaning that

$$R_m^{++}(x) = R_m^+(x) = R_m(x) \quad \text{when} \quad x < a_m.$$

• When $x \geq a_m$, $R_m^+$ is a linear function with positive slope, which is increasing. Therefore, the ReLU activation is active if and only if

$$\frac{(m+2)\|h\|_\infty}{\delta}(x - a_m) \geq (m+2)\|h\|_\infty \frac{a_{m+1} - a_m}{2\delta} \quad \Leftrightarrow \quad x \geq \frac{a_m + a_{m+1}}{2}.$$

As a result, the function $R_m^{++}$ obtained after the second operation (b) is given by:

$$R_m^{++} = \begin{cases} R_m & \text{if } x < a_m, \\ \frac{(m+2)\|h\|_\infty}{\delta}(x - a_m) & \text{if } x \in [a_m, \frac{a_m + a_{m+1}}{2}), \\ -\frac{(m+2)\|h\|_\infty}{\delta}(x - a_{m+1}) & \text{if } x \in [\frac{a_m + a_{m+1}}{2}, +\infty). \end{cases}$$

Finally, the last operation (c) cuts off the level set $\{R_m^{++} \geq (m+2)\|h\|_\infty\}$, leading to the following expression:

$$R_{m+1} = \begin{cases} R_m & \text{if } x < a_m, \\ \frac{(m+2)\|h\|_\infty}{\delta}(x - a_m) & \text{if } x \in [a_m, a_m + \delta), \\ (m+2)\|h\|_\infty & \text{if } x \in [a_m + \delta, a_{m+1} - \delta), \\ -\frac{(m+2)\|h\|_\infty}{\delta}(x - a_{m+1}) & \text{if } x \in [a_{m+1} - \delta, +\infty). \end{cases}$$

It is then easy to check that conditions C1-C5 hold, which enrolls the induction. Thus, we are able to build an increasing trapezoid function $R_M$ supported on all the $M$ subdivisions of $h$. Before moving on, we remark that $R_M$ goes to $-\infty$ as $x \to \infty$. This negative tail can be easily removed by a max operator:

$$R_M^* = \max\{R_M, 0\},$$

which sets all the negative values to zero. The obtained function $R_M^*$ is the desired increasing trapezoid function as shown in Figure 10.

**Adjusting the function values on each subdivision.** Now the goal is to approximate $h$ given the increasing trapezoid function $R_M^*$. The target function $h$ may take arbitrary function values on each subdivision while $R_M^*$ is always increasing. The idea is to properly adjust the level set of $R_M^*$ based on its increasing property. More concretely, we define its level set $L_k$ by

$$L_k = \{x \mid k\|h\|_\infty < R_M^* \leq (k+1)\|h\|_\infty\} \quad \text{for any } k = 0, \cdots, M. \tag{3}$$

By construction, the $\delta$-interior of the $k$-th subdivision $I_k^\delta$ belongs to the $k$-th level set $L_k$, i.e. $I_k^\delta \subset L_k$ for any $k \geq 1$. Then, the operations applied to the level set $L_k$ automatically transfer to operations applied to subdivision $I_k^\delta$.

For any $k = M, \cdots, 1$, we sequentially construct $R_{k-1}^*$ with

$$R_{k-1}^* = R_k^* + \frac{h_k - (k+1)\|h\|_\infty}{\|h\|_\infty}[R_k^* - k\|h\|_\infty]_+. \tag{4}$$

We prove by induction that the operation from $R_k^*$ to $R_{k-1}^*$ successfully adjusts the $k$-th subdivision $I_k^\delta$ to the desired value $h_k$. In particular, for any $k = M, \cdots, 0$, we show that $R_k^*$ satisfies

(a) $R_k^* = 0$ on $(-\infty, a_0]$ and $[a_M, +\infty)$;

(b) $R_k^* = h_j$ on $I_j^\delta$ for any $j = M, \cdots, k+1$;

(c) $R_k^* = (j+1)\|h\|_\infty$ on $I_j^\delta$ for any $j = k, \cdots, 1$;

(d) $R_k^*$ is bounded with $-\|h\|_\infty \leq R_k^* \leq (k+1)\|h\|_\infty$.

Figure 13: An illustration of the function adjustment procedure applied to the top level sets.

**Induction on $R_k^*$.** First, it is easy to check that $R_M^*$ satisfies properties (a)-(d). By induction, we assume that these properties are satisfied by $R_k^*$, and we show that they still hold for $R_{k-1}^*$. From (4), the activation function is inactive when

$$R_k^* \leq k\|h\|_\infty.$$

We show that this is the case for any $x \in (-\infty, a_0] \cup [a_M, +\infty) \cup I_M^\delta \cdots \cup I_{k+1}^\delta \cup I_{k-1}^\delta \cup \cdots \cup I_1^\delta$:

- If $x \in (-\infty, a_0] \cup [a_M, +\infty)$, from property (a), $R_k^* = 0 \leq k\|h\|_\infty$.

- If $x \in I_j^\delta$ for some $j \in M, \cdots, k+1$, from property (b), $R_k^* = h_j \leq \|h\|_\infty \leq k\|h\|_\infty$.

- If $x \in I_j^\delta$ for some $j \in k-1, \cdots, 1$, from property (c), $R_k^* = (j+1)\|h\|_\infty \leq k\|h\|_\infty$.

Therefore, we have

$$R_{k-1}^* = R_k^* \quad \text{when} \quad x \in (-\infty, a_0] \cup [a_M, +\infty) \cup I_M^\delta \cdots \cup I_{k+1}^\delta \cup I_{k-1}^\delta \cup \cdots \cup I_1^\delta.$$

This implies $R_{k-1}^*$ satisfies (a) and (c). Moreover, property (b) follows directly from (4) since it adjusts the level set $\{R_k^* \geq k\|h\|_\infty\}$ to value $h_k$. Finally, property (d) holds by remarking that

$$R_k^* \in [k\|h\|_\infty, (k+1)\|h\|_\infty] \implies R_{k-1}^* \in [-\|h\|_\infty, k\|h\|_\infty],$$

which completes the induction.

The final step is to remark that the obtained $R_0^*$ is the desired approximation of $h$. More precisely, we have shown that

- $R_0^* = 0$ on $(-\infty, a_0]$ and $[a_M, +\infty)$.

- $R_0^* = h_k$ on $I_k^\delta = [a_{k-1} + \delta, a_k - \delta]$ for any $k = 1, \cdots, M$.

- $R_0^*$ is bounded with $-\|h\|_\infty \leq R_0^* \leq \|h\|_\infty$.

As a result, the difference between $R_0^*$ and $h$ can be bounded by

$$\int_{\mathbb{R}} |R_0^*(x) - h(x)| dx \leq 4M\delta \|h\|_\infty,$$

which can be made arbitrarily small by choosing an appropriate $\delta$. This completes the proof. $\square$

The function adjustment procedure is based on the property of separated level sets, which can be applied even if the function is not an increasing trapezoid function. This is essential for dealing with the higher dimensional case since the concept of monotonicity does not naturally generalize in high dimension. Instead, we introduce the following notion of a grid indicator function.

**Definition B.2.** *In $d$-dimensional space, a subset $I$ is a hypercube if it is the Cartesian product of $d$ bounded intervals, i.e.*

$$I = [a^1, b^1) \times [a^2, b^2) \times \cdots \times [a^d, b^d).$$

*For small enough $\delta$, we denote by $I^\delta$ the $\delta$-interior of $I$, namely*

$$I^\delta = [a^1 + \delta, b^1 - \delta) \times [a^2 + \delta, b^2 - \delta) \times \cdots \times [a^d + \delta, b^d - \delta).$$

**Definition B.3.** *We call a function $g : \mathbb{R}^d \to \mathbb{R}$ a **grid indicator function** if there exist $M$ disjoint hypercubes $(I_k)_{k=1,..,M}$ such that*

- $g(x) = 0$ *if* $x \notin \cup_{k=1}^M I_k$;

- $g(x) = g_k$ *if* $x \in I_k^\delta$, *for any* $k = 1, .. M$;

- $g_i \neq g_j$ *if* $i \neq j$.

*In other words, $g$ is constant with different function values on the interior of different hypercubes. For instance, an increasing trapezoid function is a grid indicator function when $d = 1$.*

## C   Extension to higher dimensions

We extend our proof to high dimension by following the same path as our one dimensional construction. We first construct a $d$-dimensional grid indicator function and then adjust the function value on each grid cell one after another. This last step of function adjustment is performed by sliding through all the grid cells and adjusting the function value sequentially, which is essentially the same as the one-dimensional case. Therefore, the main effort is to build a $d$-dimensional grid indicator function that enjoys the separated level set property.

A piecewise constant function $h : \mathbb{R}^d \to \mathbb{R}$ (following Definition A.1) can be represented as

$$h(x) = \sum_{k=1}^{M_{1:d}} h_k \mathbb{1}_{x \in I_k},$$

where $M_{1:d} = \prod_{i=1}^d M_i$ denotes the total number of hypercubes and each $I_k$ is a $d$-dimensional hypercube of the form

$$I_k = [a_{i_1-1}^1, a_{i_1}^1) \times [a_{i_2-1}^2, a_{i_2}^2) \times \cdots \times [a_{i_d-1}^d, a_{i_d}^d)$$

for some $i_1 \in [1, M_1], i_2 \in [1, M_2], \cdots, i_d \in [1, M_d]$. Moreover, the support of $h$ is denoted by

$$I = \cup_{k=1}^{M_{1:d}} I_k = [a_0^1, a_{M_1}^1) \times [a_0^2, a_{M_2}^2) \times \cdots \times [a_0^d, a_{M_d}^d).$$

**Proposition C.1.** *Given a piecewise constant function $h : \mathbb{R}^d \to \mathbb{R}$, for any small enough $\delta > 0$, there exists a ResNet $R$ with one-neuron hidden layers such that*

- $R(x) = 0$ *if* $x \notin I$.

- $R(x) = h_k$ *for* $x \in I_k^\delta$, *which is the $\delta$-interior of the $k$-th grid cell $I_k$.*

- $R$ *is bounded with* $-\|h\|_\infty \leq R \leq \|h\|_\infty$.

*Proof.* We perform an induction on the dimension $d$. It is shown in Section B that any one-dimensional piecewise constant function can be approximated up to arbitrary accuracy when $d = 1$. Now assume that this is true for $d - 1$, meaning that we can approximate any $(d - 1)$-dimensional piecewise constant function.

Given a $d$-dimensional piecewise constant function $h$, we first decompose its support into a product of one-dimensional intervals and $(d - 1)$-dimensional hypercubes. More precisely, we write

$$J_i = [a_{i-1}^1, a_i^1) \quad \text{for } i = 1 \cdots M_1;$$
$$K_l = [a_{i_2-1}^2, a_{i_2}^2) \times \cdots \times [a_{i_d-1}^d, a_{i_d}^d) \quad \text{for } i_2 \in [1, M_2], \cdots, i_d \in [1, M_d].$$

Therefore each $I_k$ can be represented by $J_i \times K_l$, for some $i \in [1, M_1]$ and $l \in [1 : M_{2:d}]$. Next, we construct a $(d - 1)$-dimensional grid indicator function and a one-dimensional grid indicator function matching the supports $K_l$ and $J_i$ respectively.

By induction, there is a $(d - 1)$-dimensional ResNet $R_{d-1}$ such that

- $R_{d-1}(x_{2:d}) = 0$ if $x_{2:d} \notin K = \cup K_l$

- $R_{d-1}(x_{2:d}) = (l + 1)\|h\|_\infty$ for $x_{2:d} \in K_l^\delta$.

- $R_{d-1}$ is bounded with $-(M_{2:d} + 1)\|h\|_\infty \leq R_{d-1} \leq (M_{2:d} + 1)\|h\|_\infty$.

We use the notation $x_{2:d}$ to denote a $d - 1$-dimensional vector. Even though $R_{d-1}$ is $(d - 1)$-dimensional, we can extend it to a $d$-dimensional network by setting all the weights of the first coordinate to zero, as shown in Figure 14.

Figure 14: Extension of a $(d - 1)$-dimensional ResNet $R_{d-1}$ to a $d$-dimensional ResNet by setting all the weights of the first coordinate to zero.

Next, we construct an increasing trapezoid function $R_1$ on the first coordinate $x_1$ such that

- $R_1(x_1) = 0$ outside $J = \cup J_i$;

- $R_1$ is a trapezoid function on each $J_i$, for $i = 1 \cdots M_1$;

- $R_1(x_1) = \left(M_{2;d} + 1 + \frac{i}{M_1+1}\right)\|h\|_\infty$ for $x_1 \in J_i^\delta$;

- $R_1$ is bounded with $0 \leq R_1 \leq (M_{2:d} + 2)\|h\|_\infty$.

We concatenate $R_1$ with $R_{d-1}$ into a $d$-dimensional ResNet. This is possible since $R_1$ only operates on the first coordinate while $R_{d-1}$ operates on the last $d - 1$ coordinates, see Figure 15.

Figure 15: Concatenation of $R_1$ and $R_{d-1}$ in a $d$ dimension network.

Thanks to the identity mapping, we can pass the information forward even though the weights of the hidden layers are set to zero. Thus, in the last layer of the above ResNet, we get $R_1(x_1)$ in the first neuron and $R_{d-1}(x_{2:d})$ in one of the last $d-1$ neurons. Now we are going to couple these two neurons by summing them up. For technical reasons, we need to ensure the positiveness of $R_{d-1}$, which can be easily obtained by performing a max operator

$$R_{d-1}^+ = \max\{R_{d-1}, 0\}.$$

Then we sum up $R_1$ and $R_{d-1}^+$ by performing

$$R_1^+ = R_1 + [R_{d-1}^+]_+ = R_1 + R_{d-1}^+.$$

Figure 16: Summing up one-dimensional grid indicator functions on the first (left) and second (middle) coordinate. Both functions can be constructed independently by our one hidden unit ResNet.

The summation gives the desired shape inside each grid cell. However, some regions outside the grid cells are also raised, but were supposed to be zero. We show that $R_1^+$ enjoys a separated level set property which allows us to scale the outside to zero again. More precisely,

(a) When $x \notin I$, one of the function $R_1, R_{d-1}^+$ vanishes, thus

$$R_1^+(x) \le \max\{R_1, R_{d-1}^+\} \le (M_{2:d} + 2)\|h\|_\infty.$$

(b) When $x \in I_k^\delta = J_i^\delta \times K_l^\delta$, then

$$R_1^+(x) = R_1(x_1) + R_{d-1}(x_{2:d})$$
$$= \left( M_{2;d} + 1 + \frac{i}{M_1 + 1} \right) \|h\|_\infty + (l+1)\|h\|_\infty$$
$$> (M_{2:d} + 3)\|h\|_\infty. \quad \text{(since } i \geq 1 \text{ and } l \geq 1).$$

As a result, we can reset $\mathbb{R}^d \backslash I$ to zero by operating the level sets $\{R_1^+ \leq (M_{2:d} + 2)\|h\|_\infty\}$:

$$R_1^{++} = \max\{R_1^+, (M_{2:d} + 2)\|h\|_\infty\}.$$
$$R_1^* = R_1^{++} - (M_{2:d} + 2)\|h\|_\infty.$$

Then, we obtain a $d$-dimensional function $R_1^*$ with

- $R_1^* = 0$ if $x \notin I$.

- $R_1^* = \left( l + \frac{i}{M_1+1} \right) \|h\|_\infty$ on $J_i^\delta \times K_l^\delta$.

- $R_1^*$ is bounded with $0 \leq R_1^* \leq (M_{2:d} + 1)\|h\|_\infty$.

In particular, different pairs $(i, l)$ yield different values of $R_1^*$. Therefore $R_1^*$ is a $d$-dimensional grid indicator function of the desired hypercube $I$. Then it suffices to perform the function adjustment procedure on each individual grid cell to obtain the final approximation, as in the one-dimensional case. This completes the proof. □

## D    Experimental settings

In this section, we provide more details of the experimental setting for the unit ball classification problem.

**Training set.**    The training/testing samples are 2-dimensional vectors. We classify $x$ as a positive sample if $\|x\|_2 \leq 1$ and $x$ as a negative sample sample if $2 \leq \|x\|_2 \leq 3$. The training set contains $10^2$ positive samples and $2 * 10^2$ negative samples, being randomly generated.

Figure 17: A five layer fully connected network with width $d = 2$.

**Training algorithm.**    We train the network with logistic loss using SGD with momentum. We run the algorithm for 10 epochs and we observe that after 5-8 epochs the loss on the training set saturates.

**Visualizing the decision boundaries.**    After training, the learned neural network model provides a classification function $f_\mathcal{N}$. We assign positive predictions to $\{f_\mathcal{N} > 0\}$ and negative predictions to $\{f_\mathcal{N} \leq 0\}$. To visualize the decision boundary, we randomly sampled $2 * 10^3$ points in the ball $B(0, 5)$ and use red point to represent positive predictions and blue points to represent negative predictions.

## E    Proof of Proposition 2.1

We recall Proposition 2.1 in the main paper and prove it based on the result developed in [21].

**Proposition E.1.** *Let $f_\mathcal{N} : \mathbb{R}^d \to \mathbb{R}$ be the function defined by a fully connected network $\mathcal{N}$ with ReLU activation. Denote by $P = \left\{x \in \mathbb{R}^d \,|\, f_\mathcal{N}(x) > 0\right\}$ the positive level set of $f_\mathcal{N}$. If each hidden layer of $\mathcal{N}$ has at most $d$ neurons, then*

$$\lambda(P) = 0 \quad or \quad \lambda(P) = +\infty, \quad where \ \lambda \ denotes \ the \ Lebesgue \ measure.$$

*In other words, the non-trivial level set of a narrow fully connected network is always unbounded.*

*Proof.* When $d = 1$, the function $f_\mathcal{N}$ defined by a fully connected network with one hidden unit per layer is always monotone. Thus the statement holds.

When $d \geq 2$. We apply Lemma 1 of [21]: if a fully connected network $\mathcal{N}$ with ReLU activation has at most $d$ neurons per hidden layer, then

$$\int_{\mathbb{R}^d} |f_\mathcal{N}(x)| dx = 0 \ \text{ or } +\infty.$$

It is clear that $\int_{\mathbb{R}^d} |f_\mathcal{N}(x)| dx = 0$ implies $\lambda(P) = 0$. Thus it remains to consider the infinite case. However, we cannot directly obtain $\lambda(P) = \infty$, since the infinite $\ell_1$ mass may be due to the negative part of $f_\mathcal{N}$.

We are going to stack one more layer on top of $\mathcal{N}$ to build a new network $\mathcal{N}^+$ which removes its negative part.

Figure 18: Extending $\mathcal{N}$ to $\mathcal{N}^+$.

More precisely, we take the exact same coefficients as $\mathcal{N}$ and duplicate the last linear transformation into two ReLU activation functions such that

$$f_{\mathcal{N}^+} = \text{ReLU}(f_\mathcal{N}) - \text{ReLU}(f_\mathcal{N} - 1) = \begin{cases} 1 & \text{if } f_\mathcal{N} \geq 1; \\ f_\mathcal{N} & \text{if } f_\mathcal{N} \in (0, 1]; \\ 0 & \text{if } f_\mathcal{N} \leq 0. \end{cases}$$

Since $\mathcal{N}^+$ is also a fully connected network with at most $d$ neurons per hidden layer, Lemma 1 of [21] also applies to $\mathcal{N}^+$. Therefore,

$$\int_{\mathbb{R}^d} |f_{\mathcal{N}^+}(x)| dx = \int_{\mathbb{R}^d} f_{\mathcal{N}^+}(x) dx = 0 \ \text{ or } +\infty$$

Again the case when it is zero directly implies $\lambda(P) = 0$. Moreover, $f_{\mathcal{N}^+}$ is upper bounded by one, which yields

$$\int_{\mathbb{R}^d} f_{\mathcal{N}^+}(x) dx \leq \int_{\mathbb{R}^d} \mathbb{1}_{f_\mathcal{N} > 0} dx = \lambda(P).$$

Therefore, $\int_{\mathbb{R}^d} |f_{\mathcal{N}^+}(x)| dx = \infty$ implies $\lambda(P) = \infty$, which concludes the proof. $\square$

## F  Representing one-hidden unit ResNet by $d+1$ fully connected networks

In this section, we address the question about how a width $d+1$ fully connected network can represent a one-hidden unit ResNet. This implies that width-$(d + 1)$ fully connected networks are universal approximators, which is the minimum width needed [11].

First, we remark that there are two types of connections in the ResNet: ReLU activations and linear transformations (including the identity mapping), whereas in fully connected networks, ReLU is applied to every neuron. In general, applying ReLU after a linear transformation modifies the output since ReLU only takes positive values. However, in the special case that the linear transformation is always positive, adding a ReLU activation will have no effect. The main idea of our construction is to introduce a preprocessing step which projects the domain to a compact hypercube $I$. This projection step does not affect the output of the network but it allows us to represent any linear operator $\mathcal{L}$ by a shifted ReLU function:

$$\mathcal{L}(x) = [\mathcal{L}(x) - a]_+ + a \quad \text{for any } x \in I \text{ where } a = \min_{x \in I} \mathcal{L}(x). \tag{5}$$

Then we can easily construct a fully connected network matching the given ResNet by adjusting the bias coefficients.

Figure 19: Construction of a width-$(d+1)$ fully connected network based on a one-hidden unit ResNet. ReLU activation is applied to each of the green nodes, whereas blue nodes only perform linear transformations.

We first introduce the projection step. In fact, any ResNet we constructed has compact support since it belongs to $\ell_1(\mathbb{R}^d)$. Therefore, given a ResNet $R$ there is $N > 0$ such that $R$ vanishes outside the hypercube $I = [-N, N]^d$. In particular,

$$R(x) = R(\mathcal{P}(x)),$$

where $\mathcal{P}(x)$ is the coordinate-wise projection to the interval $[-N, N]$. This relationship holds because $\mathcal{P}(x)$ projects all the points outside $I$ to its boundary and, by continuity, $R$ vanishes on the boundary of $I$. Therefore, the ResNet $R$ is now applied to the compact domain $I$ which allows us to apply (5) to build a $d+1$ fully connected network $\mathcal{F}$. The construction is shown in Figure 19, where two consecutive layers of the ResNet is compressed into one layer of the fully connected network. More precisely, we construct by induction, layer by layer, a network $\mathcal{F}$ such that:

- For any ReLU neuron in the ResNet, such as $X_0$, $Y_0$, we build a corresponding neuron $\tilde{X}_0$, $\tilde{Y}_0$ in the fully connected network maintaining its value, i.e. $\tilde{X}_0 = X_0$, $\tilde{Y}_0 = Y_0$ on $I$.

- For any linear neuron in the ResNet, such as $X_1, \cdots, X_d$, we build a corresponding neuron $\tilde{X}_i$ in the fully connected network with a shifted ReLU, meaning that $\tilde{X}_i = X_i + a_i$ for some $a_i \in \mathbb{R}$ such that $\tilde{X}_i \geq 0$ on $I$.

**Technical part: induction step.** Given the $\tilde{X}_i$, we construct the $\tilde{Y}_i$. In the ResNet, $Y_i$ has the expression

$$Y_0 = \left[ \sum_{i=1}^d \beta_i Y_i + \gamma_i \right]_+ \quad \text{and} \quad Y_i = X_i + \alpha_i X_0.$$

For the induction, we now construct $\tilde{Y}_i$ from $\tilde{X}_i$ as

$$\tilde{Y}_0 = \left[ \left( \sum_{i=1}^d \alpha_i \beta_i \right) \tilde{X}_0 + \sum_{i=1}^d \beta_i \tilde{X}_i + \gamma_i - \sum_{i=1}^d \beta_i a_i \right]_+$$

and

$$\tilde{Y}_i = \left[ \tilde{X}_i + \alpha_i \tilde{X}_0 - b_i \right]_+ \quad \text{with} \quad b_i = \min_{x \in I} \{ \tilde{X}_i + \alpha_i \tilde{X}_0 \}.$$

Here, for the ReLU neurons, we appropriately adjust the bias to maintain the function value. Moreover, the output layer, which is a linear transformation, can also be maintained in the same way as $Y_0$. As a consequence, the fully connected network $\mathcal{F}$ matches $R$ on $I$, leading to

$$\mathcal{F}(\mathcal{P}(x)) = R(\mathcal{P}(x)) = R(x).$$

Finally, we represent the projection by a shifted fully connected network, and concatenate $\mathcal{F}$ to it:

$$\mathcal{P}(x_i) = \left[2N - [N - x_i]_+\right]_+ - N, \quad \forall i \in [1, d]. \tag{6}$$

**Conclusion:** Our construction shows that ResNet can be represented by a specific fully connected network, and hence implies the same representation power as ResNet for this fully connected network with width $d + 1$. But the proof does not imply the other direction in general. In particular, the constructed fully connected network is very *sparse*: it has $O(d)$ connections instead of the $O(d^2)$ connections that fully connected networks generally have.