[Reviews · NeurIPS 2018]

Reviewer 1



The authors proved the universal approximation theorem for the ResNet architecture. It turned out that thanks to the special architecture ResNets are universal approximators opposed to fully connected networks for which universal approximation fails with width d or less. Conclusion: 1) the topic of the paper is very important 2) the paper is well written. It almost does not contain misprints except - line 230. “networks may be a patrial explanation…” 3) the results close the existing gap in the theory: it follows that fully connected networks with width (d+1) are universal approximators while width d fully connected networks are not; the previous construction required at least d+4 units per layer. 4) There is nothing to add to the review, at least from my side.

Reviewer 2



Universal approximation theoresm have been of interest in Mathematics and, more recently, in Neural Networks. There are different classes of approximators: polynomials, wide neural neworks, deep neural networks, etc. The paper is the first to show that width-limited ResNets are universal approximators. This has been an open problem.

Reviewer 3



In this paper, the authors demonstrated the universal approximation of ResNet with one-neuron hidden layer. In more details, ResNet with one single neural per hidden layer can approximate any Lebesgue-integrable functions to any accuracy as the depth goes to infinity. This also shows that ResNet has more representation power than narrow deep networks which are not universal approximators if the width is the input dimension. This result seems interesting since universal approximation is a fundamental property for any models. However, my comment may not be correct since I am not familiar with approximation theory of neural networks. I also did not check the involved proof. The authors shows that the universal approximation of ResNet with one-neuron hidden layer implies the width-d+1 fully connected network are universal approximators. I did not fully understand this. From my understanding, it seems that ResNet with one-neuron hidden layers can not be understood as width-d+1 fully connected networks since there is no activation function applied to the identity map. Furthermore, there is a linear transformation applied to the activation function in the ResNet architecture. After rebuttal: The authors' response addresses my concerns. I am in favor of its acceptance.